# The Contribution of Melanoidins to Soy Sauce Antioxidant Activities and Their Structure Characteristics

**DOI:** 10.3390/foods14162787

**Published:** 2025-08-11

**Authors:** Hanhan Li, Yaqiong Zhang, Zhi-Hong Zhang, Feng Wang, Baoguo Xu, Zhankai Zhang, Haile Ma, Xianli Gao

**Affiliations:** School of Food & Biological Engineering, Jiangsu University, Zhenjiang 212013, China; 18337010038@163.com (H.L.); 2222018115@stmail.ujs.edu.cn (Y.Z.); zhihong1942@ujs.edu.cn (Z.-H.Z.); fengwang@ujs.edu.cn (F.W.); xbg@ujs.edu.cn (B.X.); zhangzhankai1998@sina.com (Z.Z.); mhl@ujs.edu.cn (H.M.)

**Keywords:** Maillard reaction, soy sauce melanoidins, molecular weight, bioactivity, antioxidant contributions, functional food

## Abstract

Melanoidins, generated during the Maillard reaction in soy sauce fermentation, have potential health benefits due to its excellent bioactivity. This study aimed to investigate the antioxidant contributions and structural characteristics of melanoidins in soy sauce. Five molecular weight fractions (1–3 kDa, 3–10 kDa, 10–30 kDa, 30–50 kDa, and >50 kDa) were isolated and their composition was analyzed. Results showed that soy sauce melanoidins mainly comprised proteins, sugars, and phenolic compounds. Antioxidant activities of the melanoidins were influenced by their molecular weights and structures. The >50 kDa melanoidins fraction contributed the most to the overall antioxidant activities of soy sauce. The total contributions of melanoidins to the antioxidant activities of soy sauce ranged from 34.21% to 75.03%. Spectroscopic analyses indicated that the antioxidant activities were positively correlated with the presence of conjugated structures and active functional groups (i.e., C=C, C=O, N-H, O-H) in melanoidins. This study provides new insights into the health-promoting properties of soy sauce melanoidins and offers theoretical support for the development of soy sauce as a functional food.

## 1. Introduction

Soy sauce, a traditional fermented condiment known for its distinctive flavor and color, plays a key role in Asian foods [1,2]. Soy sauce production includes two key stages, Koji fermentation and moromi fermentation. During the koji fermentation, *Aspergillus oryzae* is inoculated into steamed soybeans and fermented for approximately 44 to 48 h to make koji with abundant enzymes [3]. In the subsequent moromi fermentation, the mash is placed outdoors to ferment in an 18–20% (*w*/*v*) brine for six months at the temperatures that fluctuate between 20 °C and 40 °C. The condition promotes the Maillard reaction and other biochemical transformations that contribute to the characteristic flavor and color of soy sauce [2,3,4]. In this process, the Maillard reaction plays a crucial role by generating Maillard reaction products, with melanoidins are the primary components [4,5]. The formation of melanoidins in foods is significantly influenced by factors such as ingredient composition, temperature, pH value, and fermentation time [6,7,8]. The molecular weights and structures of melanoidins in fermented foods vary with fermentation time, leading to changes in the physicochemical properties of melanoidins [9]. However, the detailed molecular weights and structural characteristics of melanoidins in soy sauce remain unclear till now.

Recent studies have highlighted melanoidins as multifunctional bioactive compounds with antioxidant, prebiotic, and anti-inflammatory properties [10,11,12]. Among these properties, their antioxidant capacity has received increasing attention, especially in food systems. In thermally processed foods such as coffee, bread, and cocoa, melanoidins exhibit significant antioxidant activity through free radical scavenging and metal chelation, which help extend the shelf life of these products [13,14]. Similarly, melanoidins provide various health benefits through their antioxidant action in traditional fermented foods like soy sauce, beer, vinegar, and tempeh [13]. For instance, melanoidins in vinegar contribute significantly to its antioxidant capacity, accounting for approximately 50% of the antioxidant activity in Zhenjiang aromatic Vinegar [15]. High molecular weight melanoidins in beer can protect cells from oxidative DNA damage and inhibit lipid peroxidation through metal chelation compared to low molecular weight melanoidins [16,17]. Notably, soy sauce, a common edible condiment with a per capita intake of 10–20 g per day, particularly among Asian populations, exhibits significantly higher free radical scavenging capacity than several other fermented products, including vinegar, cooking wine, and yellow wine [4,18]. Dark soy sauce, in particular, shows antioxidant activity approximately ten times that of red wine [19]. The antioxidant properties of soy sauce are primarily attributed to melanoidins, which are formed through non-enzymatic browning reactions during fermentation. Previous studies have demonstrated that soy sauce melanoidins effectively neutralize free radicals, thereby inhibiting DNA damage induced by nitric oxide (NO) [20]. As the primary contributors to soy sauce’s antioxidant capacity, the content of melanoidins increase in concentration with the duration of soy sauce fermentation. [4] Studies indicate that dark soy sauce contains higher levels of melanoidins (3–55 kDa), with a concentration of 22.8 g/100 mL [21]. The studies above indicate that soy sauce melanoidins play a critical role in maintaining the antioxidant activities of foods. However, the specific contributions of pure melanoidins to the overall antioxidant activities of various food systems have not been systematically and thoroughly investigated. Additionally, the relationship between the molecular weight, structure, and antioxidant potential of melanoidins in soy sauce remains unclear. This research gap may seriously hinder the development and utilization of melanoidins as a functional ingredient.

Therefore, this study aims to (i) prepare soy sauce melanoidins with varying molecular weights, (ii) evaluate their specific contributions to the antioxidant activities of soy sauce, and (iii) explore the structural characteristics of melanoidins and their relationship with the antioxidant properties of soy sauce.

## 2. Materials and Methods

### 2.1. Materials

The DA201-E macroporous adsorbent resins were purchased from Zhengzhou Hecheng New Materials Technology Co., Ltd. (Zhengzhou, China). The ultrafiltration centrifugation tubes (1 kDa, 3 kDa, 10 kDa, 30 kDa, 50 kDa 20 mL) were purchased from Pall Corporation (Boston, Massachusetts, USA). All the chemicals and solvents were of analytical grade and were purchased from Sinopharm Chemical Reagent Co., Ltd. (Shanghai, China).

### 2.2. Extraction of Soy Sauce Melanoidins from Soy Sauce

#### 2.2.1. Extraction of Melanoidins by Macroporous Resin Adsorption

Soy sauce was produced following the method described by Gao et al. [2] which was fermented for six months and utilized in the subsequent experiments. The melanoidins were prepared using resin according to the method outlined by Wang et al. [22] Briefly, 200 g of DA201-E resin was soaked in analytical ethanol for 24 h, followed by washing with distilled water to remove the residual impurities. Next, 100 mL of soy sauce was diluted with distilled water, adjusting its absorbance at 405 nm to 1.00 ± 0.02. The diluted soy sauce solution was then passed through a column packed with pretreated DA201-E resin (inner diameter: 2.5 cm, length: 60 cm) at a flow rate of 2 mL/min. Once adsorption equilibrium was achieved, the resin was eluted with 200 mL of 40% (*v*/*v*) ethanol (equivalent to 4 bed volumes) at a flow rate of 3 mL/min. The ethanol in the eluate was subsequently evaporated using a rotary evaporator (Eyela N-1001, Tokyo, Japan) at 55 °C and a resultant brown eluate was obtained.

#### 2.2.2. Ultrafiltration Separation of the Extracted Melanoidins

The resultant brown eluate was filtered using ultrafiltration tubes with molecular weight cut-off ranges of 50, 30, 10, 3, and 1 kDa to prepare melanoidins fractions with molecular weight. Initially, the brown eluate was centrifuged at 4000× *g* for 30 min at 4 °C through a 50 kDa filter to separate the >50 kDa fraction. The filtrate was then sequentially passed through 30, 10, 3, and 1 kDa filters under the same centrifugation conditions, yielding fractions of 30–50 kDa, 10–30 kDa, 3–10 kDa, and 1–3 kDa, respectively. Each retained fraction was collected, and the final permeate represented the <1 kDa fraction. All melanoidins fractions and the resulting brown eluate were subsequently lyophilized using a DELTA 1-24LSC Christ freeze dryer (Martin Christ, Osterode am Harz, Germany) and stored in desiccators for further analyses.

### 2.3. Determination of the Melanoidins Content

The content of melanoidins was determined using a previous study with slight modifications [23,24] and the melanoidins content was calculated using Equation (1):
(1)$$C=\frac{A\times V}{0.269\times V_{0}\times 10}$$
where *C* indicates the content of melanoidins (mg/mL), *A* indicates the absorbance value (470_nm_) of the sample solution, *V* indicates the volume of the fixed volume (mL), *V*_0_ indicates the volume of the sample (mL), and 0.269 indicates the absorbance value of 1.0 mL of aqueous solution containing 0.1 mg of melanoidins.

### 2.4. Determination of Chemical Composition of Melanoidins

#### 2.4.1. Total Sugar, Protein, and Total Phenol Content

The total sugar, protein, and total phenol content of melanoidins were measured using the phenol–sulfuric acid method [25], the Bradford method, and the Folin–Ciocalteu method [26], respectively. Standard curves were constructed using glucose for total sugars, bovine serum albumin (BSA) for protein, and gallic acid (GAE) for total phenols. The results were expressed as the content of each chemical constituent per gram of dry weight of melanoidins.

#### 2.4.2. Free Amino Acids Composition

The Amino acids in melanoidins were quantified using the method described by Gao et al. [3] Briefly, 0.1 g of the melanoidins sample was mixed with 2.5 mL of 5% trichloroacetic acid and sonicated for 20 min at room temperature. The mixture was then centrifuged at 15,000× *g* for 30 min. One milliliter of the supernatant was collected, filtered through a 0.22 μm membrane, and transferred to a sample vial. Amino acids separation was performed on a column (250 × 4.6 mm, 5 μm; Xtimate C18) by HPLC system (LC-20AD, Shimazu, Tokyo, Japan) at 40 °C. The mobile phase consisted of (A) 0.1% (*v*/*v*) trifluoroacetic acid (TFA) in water and (B) 0.1% (*v*/*v*) TFA in acetonitrile. The gradient elution program was as follows: 0–2 min, 100% A; 2–20 min, linear gradient to 60% A and 40% B; 20–25 min, linear gradient to 100% B; 25–30 min, 100% B; followed by re-equilibration to 100% A for 10 min. The wavelength of detection was 254 nm, the injection volume was 10 μL, and the flow rate was 1.0 mL/min. Amino acids quantification was achieved using the external standard method, with individual amino acid standards prepared in the same solvent system. The results were expressed as mg of amino acid per g of melanoidins.

### 2.5. Determination of Antioxidant Activities of Melanoidins

#### 2.5.1. DPPH Free Radical Scavenging Activity

The DPPH radical scavenging activity was determined using the method reported by Zhao et al. [27] with some modifications. In brief, 2.0 mL of melanoidins solution (20–100 μg/mL) was mixed with 2.0 mL of DPPH ethanol solution (0.2 mmol/mL). The mixture was incubated in the dark for 30 min, after which the absorbance was measured at 517 nm. *A*_0_, *A*_1_, and *A_S_* are the absorbances of the blank (sample solution was replaced with distilled water), sample, and control solutions (DPPH solution was replaced with ethanol), respectively. Ascorbic acid was used as the positive control. DPPH scavenging activity was calculated using the following Equation (2):
(2)$$DPPH\;redical\;scavenging\;activity\;\left(\%\right)=\frac{A_{0}-\left(A_{1}-A_{s}\right)}{A_{0}}\times 100\%\;$$

The dose–response curves were constructed by plotting the scavenging percentage (Equation (2)) against the melanoidins concentration (20–100 μg/mL). The IC_50_ value (concentration required to scavenge 50% of the DPPH radicals) was determined by non-linear four-parameter logistic regression. Only curves with R^2^ ≥ 0.98 were accepted. The results were expressed as the equivalent concentration of Trolox solution (μg AAE/mL).

#### 2.5.2. ABTS Free Radical Scavenging Activity

The ABTS free radical scavenging activity was assessed based on a previously reported method [28] with minor modifications. Five milliliters of 7 mmol/L ABTS solution were mixed with 1 mL of 2.45 mmol/L K_2_S_2_O_8_ solution in a tube and the mixture was allowed to stand for 12 to 16 h at room temperature in the dark. The resulting mixture was then diluted with phosphate buffer (pH 7.4) until the absorbance at 734 nm reached 0.70 ± 0.02. Subsequently, 40 μL of melanoidins solution (0.2–1.0 mg/mL) was added to 4.0 mL of the ABTS assay solution, and the reaction was allowed to proceed for 6 min. After the incubation, the absorbance was measured at 734 nm. *A_0_* and *As* were the absorbances of the control (distilled water instead of the sample solution) and sample solutions, respectively. Trolox was used as the positive control. ABTS scavenging activity was calculated according to the following Equation (3):
(3)$$ABTS\;redical\;scavenging\;activity\;\left(\%\right)=\frac{A_{0}-A_{s}}{A_{0}}\times 100\%$$

The dose–response curves were constructed by plotting the scavenging percentage (Equation (3)) against melanoidin concentration (0.2–1.0 mg/mL). The IC_50_ value (concentration required to scavenge 50% of the ABTS radicals) was determined by non-linear four-parameter logistic regression. Only curves with R^2^ ≥ 0.98 were accepted. The results were expressed as the equivalent concentration of Trolox solution (μmol TE/mL).

#### 2.5.3. Oxygen Radical Absorbing Capacity (ORAC)

The ORAC was determined using fluorescein as a fluorescent probe. In brief, 50 μL of melanoidins solution (0.2–1.0 mg/mL), 50 μL of sodium fluorescein solution (0.096 μmol/L), and 100 μL of 119.4 mmol/L 2,2′-Azobis (2-amidinopropane) dihydrochloride solution were added to 96-well plates (Corning Scientific, Corning, NY, USA). The mixture was incubated for 30 min at 37 °C in the dark. Fluorescence measurements were then performed using a Synergy H1 Hybrid Multimode microplate reader (Biotek, Winooski, VT, USA), with the excitation wavelength set to 485 ± 20 nm and the emission wavelength set to 520 ± 20 nm. Fluorescence was recorded every 2 min over a total period of 90 min. A standard curve, y = −0.1757× + 0.7737 (R^2^ = 0.9994), was constructed using Trolox as the standard. A freshly prepared AAPH solution was utilized for each test. Trolox equivalents were calculated based on the relative area under the curve for the samples, compared with a Trolox standard curve prepared under the same experimental conditions. The results were expressed as the equivalent μmol of Trolox per gram of melanoidins (μmol TE/g).

#### 2.5.4. Metal Chelating Ability (MCA)

The MCA was determined using the method described by Wu et al. [29] One milliliter of the melanoidins solution (0.2–1.0 mg/mL) was mixed with 100 μL of 2 mmol/L FeCl_2_ solution, followed by the addition of 200 μL of 5 mmol/L phenanthroline solution. The mixture was allowed to react for 5 min at room temperature. Distilled water was then added to achieve a total volume of 6.0 mL, and the reaction was allowed to continue for an additional 10 min at room temperature. The absorbance of the resulting solution was measured at 562 nm. A standard curve, y = −0.0087× + 0.6942 (R^2^ = 0.9995), was constructed using EDTA as the standard. The results were expressed as the equivalent μg of EDTA per milligram of melanoidins (μg EE/mg).

#### 2.5.5. Ferric Reducing Antioxidant Power (FRAP)

The FRAP of melanoidins was determined as a modified version of the method [30]. In brief, 2.0 mL of melanoidins solution (0.2–1.0 mg/mL) was mixed with 2.5 mL of phosphate buffer (0.2 mol/L, pH 6.6) and 2.5 mL of potassium ferricyanide solution (1%). The mixture was incubated in a 50 °C water bath for 20 min. Then 2.5 mL of trichloroacetic acid (10%, *w*/*v*) was added, and the mixture was kept at room temperature for 10 min. Afterward, 2.5 mL of the supernatant was taken, and 2.5 mL of distilled water and 0.5 mL of ferric chloride solution (0.1%, *w*/*v*) were added. The mixture was allowed to react at room temperature for 10 min before measuring the absorbance at 700 nm. A standard curve, y = 0.0108× + 0.0322 (R^2^ = 0.9994), was constructed using ascorbic acid as the standard. The results were expressed as the equivalent μg of ascorbic acid per milligram of melanoidins (μg AAE/mg).

### 2.6. Contributions of the Antioxidant Activities of Soy Sauce Melanoidins

For the melanoidins solution preparation, each melanoidins fraction intercepted from 100 mL of soy sauce as described in Section 2.3 was dissolved in 18% brine, which was diluted to 100 mL using 18% brine. Antioxidant activities of the melanoidins fraction solution and soy sauce were assessed using the methods described in Section 2.5 (DPPH, FRAP, ABTS, ORAC, and MCA). The percentage contribution of each melanoidins fraction to the overall antioxidant activity of soy sauce was calculated using Equation (4):
(4)$$W_{i}\;\left(\%\right)=\frac{M_{i}}{N}\times 100\%\;$$ where *M_i_* is the antioxidant activities of melanoidins fraction_i_ solution (i = 1–3 kDa, 3–10 kDa, 10–30 kDa, 30–50 kDa, and >50 kDa); *N* is the antioxidant activity of soy sauce measured under the same measurement conditions, *W_i_* is the percentage contribution of fraction_i_ to the antioxidant activity of soy sauce.

### 2.7. Structural Characteristics of Melanoidins

#### 2.7.1. UV-Vis Spectra

Melanoidins were diluted with distilled water to achieve a concentration of 1.0 mg/mL. The absorbance was measured over the wavelength range of 200 to 800 nm using a UV-vis spectrometer (UV-2550, Shimadzu, Tokyo, Japan).

#### 2.7.2. Fluorescence Spectra

The fluorescence characteristics of the melanoidins solution at a concentration of 1.0 mg/mL were measured using a fluorescence spectrometer (F-4600, Foss, Denmark). During the fluorescence spectral analysis, an excitation wavelength of 350 nm and an emission wavelength range from 300 to 600 nm were utilized.

#### 2.7.3. Fourier Transform Infrared Spectroscopy (FT-IR)

FT-IR spectra of melanoidins were obtained using a Fourier Transform Infrared spectroscopy (Bruker Corp, Tensor 27, Karlsruhe, Germany). A mixture of 100 mg potassium bromide and 1.0 mg lyophilized melanoidins powder was ground into transparent flakes for FTIR analysis. The wavenumber range was set from 4000 to 400 cm^−1^, with a resolution of 4 cm^−1^ and a total of 64 scans conducted.

#### 2.7.4. 1H Nuclear Magnetic Resonance Spectroscopy (1H NMR)

The *1H NMR* spectra of melanoidins were recorded using an NMR spectrometer (AVANCE III 600 M, Bruker, MA, USA). Twenty milligrams of melanoidins were dissolved in 0.5 mL of D_2_O, thoroughly mixed, and then transferred to an NMR tube. Data were collected by performing 32 scans at a proton frequency of 600 MHz.

### 2.8. Data Analyses

All analyses were conducted in triplicate except for spectroscopy analyses. Statistical significance amongst samples and controls was assessed using Duncan’s test, with a significance threshold set at *p* < 0.05. Data analyses were performed using SPSS version 19.0 (IBM Corporation, New York, USA). The Figures were generated using Origin 2021 (Origin Lab Corporation, Northampton, MA, USA).

## 3. Results

### 3.1. Melanoidins Prepared from Soy Sauce and Their Content

Soy sauce melanoidins and their six fractions with different molecular weights were prepared using resin adsorption and ultrafiltration. There is a clear positive correlation between the molecular weight of soy sauce melanoidins and the degree of browning (Figure 1). The fraction with a molecular weight > 10 kDa demonstrates a higher degree of browning, which is attributed to the formation of melanoidins during a stronger Maillard reaction [31]. In contrast, the fraction with molecular weights < 1 kDa can be negligible, indicating that the fraction contains little melanoidins and is therefore excluded from further analyses. As shown in Table 1, the >50 kDa fraction exhibits the highest content of 0.90 g/100 mL, followed by the 30–50 kDa fraction with a content of 0.61 g/100 mL. Together, the >50 kDa and 30–50 kDa fractions account for 61.89% of the total melanoidins. This suggests that higher molecular weight melanoidins possibly play a significant role in soy sauce [25]. The molecular weight of melanoidins is influenced by both the temperature and duration of the Maillard reaction [13]. The long fermentation time (generally from spring to autumn) and high fermentation temperature outdoors in summer (approximately 50 °C) lead to the complete Maillard reaction in Chinese soy sauce, generating more melanoidins with higher molecular weights via polymerizing amongst low molecular weight melanoidins [9].

### 3.2. Chemical Composition of Melanoidins

#### 3.2.1. The Main Composition of Melanoidins

As shown in Table 2, among the three components, protein is the most abundant component in all melanoidins fractions, followed by sugars and phenolics. This is due to the fact that soy sauce is typically fermented from protein-rich raw materials such as soybeans and wheat, which are enzymatically hydrolyzed to produce amino acids and small peptides, serving as precursors for the formation of soy sauce melanoidins [3,32]. This is the reason for the higher proportion of protein in soy sauce melanoidins compared with other foods like coffee and chocolate [13,33]. The contents of both protein and sugar increase with increasing molecular weights of soy sauce melanoidins, which is consistent with the results reported by other authors for lotus rhizomes and vinegar melanoidins [17,25]. Specifically, the protein and sugar contents in the >50 kDa fraction are significantly higher than those in the other fractions (*p* < 0.05), reaching 109.20 mg/g and 28.34 mg/g, respectively. Phenolics content is the lowest of the three components, with the highest content of only 0.88 mg/g in the 10–30 kDa fraction. Compared to the phenolics content in coffee, chocolate, and sweet wine [33,34], the phenolics content of soy sauce melanoidins is lower. This is primarily attributed to the fact that grain-derived fermented foods have lower phenolic content, resulting in melanoidins proteins with reduced cross-reactivity between gluten, sugars, and phenolic substances. Furthermore, the higher salt concentration in soy sauce (18–20% *w*/*v*) increases ionic strength and disrupts the non-covalent bonds between melanoidins and phenols [35]. Some phenolics may be degraded or converted into other compounds during soy sauce fermentation, further reducing the phenolics content. The differences in the content of these main components in soy sauce melanoidins with varying molecular weights may lead to differences in their antioxidant activities.

#### 3.2.2. Free Amino Acids of Melanoidins

In the Maillard reaction, free amino acids are covalently or non-covalently bonded to the carbohydrate backbone of melanoidins, ultimately polymerizing into high molecular weight melanoidins [36]. The amino acid compositions of the five fractions are similar, with their amino acid content inversely proportional to molecular weights (Table 3). The amino acid content of the 1–3 kDa fraction is statistically higher than that of the >50 kDa fraction (*p* < 0.05). High molecular weight melanoidins bind more tightly to amino acids, exhibit strong hydrophobicity, and are more resistant to the release and dissolution of free amino acids [37]. This may explain the inverse relationship between amino acid content and molecular weight of soy sauce melanoidins. Additionally, free amino acids are highly reactive in the synthesis of melanoidins and can directly scavenge free radicals, contributing to the antioxidant capacity of melanoidins. Among the amino acids present in soy sauce melanoidins, glutamic acid, aspartic acid, and phenylalanine are the most abundant. These amino acids enhance the antioxidant capacity of soy sauce by scavenging free radicals and reacting with oxidizing agents [22]. Similar results have been observed in other food systems with characteristics akin to melanoidins. For example, serine, alanine, and glutamic acid are primarily involved in the formation of beer melanoidins, while serine and aspartic acid are found in higher concentrations in barley melanoidins [38]. Furthermore, the purity of melanoidins can be calculated by subtracting the known components (e.g., total sugars, proteins, phenolics, and free amino acids) from the total sample. The purities of the five fractions were 84.26%, 85.83%, 84.94%, 84.16%, and 83.29%, respectively, indicating that the soy sauce melanoidins skeleton is of high purity. The high purity may facilitate the binding of active groups to the melanoidins skeleton through covalent and non-covalent interactions [17], thereby enhancing their free radical scavenging ability and exhibiting strong antioxidant activities.

### 3.3. Antioxidant Activities of Melanoidins

In assessing the antioxidant activities of plant-based foods or matrices, free radical scavenging ability, metal chelation and reducing power are critical indicators of potential antioxidant effectiveness [29]. The IC50 values for DPPH and ABTS across different molecular weights fractions are presented in Figure 2a. The IC_50_ value represents the concentration required to scavenge 50% of free radicals, and a lower IC_50_ value indicates a stronger scavenging ability. Our results reveal that the IC_50_ values for the 1–3 kDa and 3–10 kDa fractions are significantly lower than those for the other fractions (*p* < 0.05), indicating stronger free radical scavenging activity in these low molecular weights fractions. This finding contrasted with previous studies such as bread, black garlic, and Monascus vinegar [10,14,30], which suggested that high molecular weights melanoidins possess superior free radical scavenging activity, and it is hypothesized that this discrepancy may stem from differences between the chemical structure of melanoidins and molecular weight. Unlike high-temperature short-term fermentation, low-temperature long-term fermentation of soy sauce substantially slows the formation of Maillard reaction products, promoting the accumulation of low molecular weight components such as small peptides and free amino acids [6,8]. Due to their smaller molecular size, lower steric hindrance, and higher diffusion coefficient, they have an increased collision frequency with free radicals. They effectively scavenge DPPH/ABTS free radicals through the hydrogen atom transfer (HAT) mechanism, resulting in lower IC_50_ values. In contrast, the 30–50 kDa and >50 kDa fractions with complex spatial structures may experience spatial hindrance, reducing their efficiency in reacting with free radicals and leading to weaker scavenging activity [25].

As shown in Figure 2b, the ORAC values of all soy sauce melanoidins fractions exceed 40 μmol TE/g. However, as shown in Section 3.2.1, soy sauce melanoidins, which are protein-based, exhibit significantly lower ORAC values compared to sugar-based melanoidins found in foods such as coffee, chocolate, and bread crusts, which typically exhibit higher ORAC values [13,33]. Pastoriza et al. [33] have already pointed out that protein-based melanoidins usually have lower ORAC values (1–70 μmol TE/g), in contrast to sugar-based melanoidins (186–246 μmol TE/g). This difference is attributed to the greater abundance of hydrogen atom donors in sugar-based melanoidins, which enhances the antioxidant activity by donating hydrogen atoms to neutralize free radicals.

In MCA and FRAP assays (Figure 2c,d), the 30–50 kDa and >50 kDa fractions exhibit higher MCA and FRAP values. Notably, in the MCA assay, the chelation abilities of the 30–50 kDa fraction (51.46 μg EE/mg) and the >50 kDa fraction (42.01 μg EE/mg) are significantly higher than that of the 1–3 kDa fraction (31.55 μg EE/mg) (*p* < 0.05). The enhanced chelation ability is likely due to the presence of pyridones or pyranones in melanoidins, which provide chelation donors such as hydroxyl and ketone groups [9]. The high molecular weight fractions, with their complex structures, can bind more tightly to Fe^2+^, resulting in superior metal ion chelation. This finding is consistent with previous research on melanoidins derived from beer and black garlic [28,39]. Moreover, as shown in Figure 3d, the FRAP assay measures electron transfer (ET) capacity, which depends on the conjugated system to promote electron delocalization [28]. High molecular weight fractions (10–30 kDa, 30–50 kDa, >50 kDa) are rich in π-π conjugated double bonds, aromatic rings, and hydroxyl groups, driving efficient electron transfer and enhancing Fe^3+^ reduction to Fe^2+^ mediated by ET. Among these, the 10–30 kDa fraction exhibits the highest FRAP value (140.59 μg AAE/mg), which is significantly higher than that of the 1–3 kDa fraction (88.88 μg AAE/mg) and the 3–10 kDa fraction (116.39 μg AAE/mg) (*p* < 0.05).

In conclusion, this study demonstrates that the antioxidant activities of soy sauce melanoidins are closely related to their molecular weights. The low molecular weights fractions exhibit stronger free radical scavenging ability, while the high molecular weights fractions show greater metal chelation and reducing power. The distinct antioxidant mechanisms of soy sauce melanoidins with varying molecular weights suggest that their antioxidant activities are strongly influenced by their structural characteristics.

### 3.4. Contributions of Melanoidins to the Antioxidant Activities of Soy Sauce

Based on the actual concentrations of five fractions in soy sauce (Table 1), corresponding melanoidins fraction solutions were prepared to assess the contribution of each fraction to the overall antioxidant activity of soy sauce. The results of the antioxidant assays, conducted using five antioxidant assays (DPPH, ABTS, ORAC, MCA, and FRAP), are presented in Table 4.

The antioxidant contribution of the >50 kDa fraction is significantly higher than other fractions (*p* < 0.05). Notably, in the MCA assay, the antioxidant contribution of the >50 kDa fraction (25.61%) is about four times greater than that the 1–3 kDa fraction (6.30%). The data in Table 4 show that high molecular weight fractions (>50 kDa and 30–50 kDa) are the primary contributors to antioxidant activities of soy sauce, accounting for 32.28%, 30.10%, and 44.19% of the antioxidant activities in the DPPH, ORAC, and MCA assays, respectively. In contrast, the antioxidant contribution of the 1–3 kDa fraction is lower, at 8.56%, 8.51%, and 6.30%, respectively. This result is consistent with the findings of Liu et al. [15], who demonstrated that, compared to low molecular weight melanoidins, high molecular weight melanoidins exhibit stronger antioxidant activities (DPPH and ORAC), contributing to about 50% of the overall antioxidant activities of Zhenjiang aromatic vinegar.

Overall, the five melanoidins fractions in this study contribute to between 34.21% and 75.03% of the total antioxidant activities of soy sauce, indicating that melanoidins is a key component of soy sauce’s antioxidant activities [13]. Additionally, studies have also shown that soy sauce contains other important antioxidant compounds, such as 4-ethylguaiacol, catechin, soybean glycoside, 4-ethylphenol, vanillin, and specific peptide segments [1,13]. These components interact synergistically with melanoidins of different molecular weights, collectively enhancing the overall antioxidant activities of soy sauce.

### 3.5. Structural Characterization of Melanoidins

#### 3.5.1. Ultraviolet–Visible Spectral Analysis

Spectral curves of the five melanoidins fractions exhibit similar UV absorption characteristics in the wavelength range of 200–800 nm (Figure 3a). In the UV region, melanoidins display both n-π* transitions of unsaturated heteroatoms and electronic π-π* transitions of unsaturated bonds [40]. As the molecular weight increases, the structure of higher molecular weight fractions becomes more complex, with stronger absorption peaks in the UV spectra. These fractions also exhibit multiple absorption peaks over a broader wavelength range, which is consistent with the UV characteristics of melanoidins [41]. For example, two distinct absorption peaks are observed at 275 nm and 325 nm, corresponding to π-π* transitions in C=C double bonds and n-π* transitions in C=O and N=O bonds [40]. The 10–30 kDa, 30–50 kDa, and >50 kDa fractions show prominent peaks at 275 nm and 325 nm, indicating the presence of more unsaturated conjugated structures (C=C, C=O) in melanoidins formed during fermentation. In contrast, spectra of the 1–3 kDa melanoidins fraction in the range of 200–380 nm do not show clear absorption peaks. This may be due to the presence of a large number of free amino acids and small peptides, whose absorption results from the superimposed effects of individual moieties rather than from a single characteristic absorption [42]. Additionally, the absorption peak at 325 nm may also indicate the presence of benzene-coupled compounds, such as flavonoids and chlorogenic acid [43]. The absorbance at 420 nm is commonly used to indicate the formation of colored melanoidins [38], which are closely associated with the chromophores of melanoidins. The 10–30 kDa, 30–50 kDa, and >50 kDa fractions exhibit higher absorbance values in the visible region compared to the 1–3 kDa and 3–10 kDa fractions, suggesting that higher molecular weight melanoidins contain more chromophores. Furthermore, as the number of chromophores increases, the energy of electronic excitation decreases, causing a red shift in the maximum absorption wavelength [34].

#### 3.5.2. Fluorescence Spectral Analysis

Fluorescent substances can be generated through the Maillard reaction [40]. As shown in Figure 3b, melanoidins in the five fractions exhibit fluorescence peaks, with maximum fluorescence intensity occurring at an excitation wavelength of 350 nm and emission wavelengths ranging from 420 to 434 nm. This suggests that soy sauce melanoidins contain conjugated structures, such as carboxyl and carbonyl groups, and similar findings have been found in previous studies [24,44]. As the molecular weight increases, the absorption peaks of fluorescence broaden, likely due to the presence of more double-bonded structural units (e.g., aromatic rings) in larger molecules. These units, along with varying electronic energy levels, enable electrons to move more freely across a broader range [44]. Furthermore, the increase in molecular weight causes π-electrons to traverse a wider range of conjugated environments, leading to changes in energy levels and a red shift in fluorescence emission wavelengths [44]. As discussed in Section 3.2.2, the 1–3 kDa and 3–10 kDa fractions contain higher concentrations of fluorescent amino acids such as tryptophan, tyrosine, and phenylalanine. This likely explains the sharper fluorescence signals observed in the 1–3 kDa and 3–10 kDa melanoidins fractions. In contrast, the reduced fluorescence intensity in the 30–50 kDa and >50 kDa fractions may be attributed to the encapsulation of fluorescent groups during melanoidin polymerization, which diminishes the fluorescence emission from these groups. Additionally, intermolecular interactions, such as π-π stacking and hydrogen bonding, may suppress fluorescence as the molecular weight increases. The highly aggregated structure of these larger molecules may lead to a fluorescence burst phenomenon [34].

#### 3.5.3. FTIR Spectral Analysis

Infrared spectroscopy is capable of identifying the primary chemical constituents of melanoidins and their structural characteristics. The spectral absorption peaks of soy sauce melanoidins are observed at 3220, 2930, 1600, 1400, 1230, 1040, and 610 cm^−1^. Infrared spectra of the five melanoidins fractions exhibit similarities in their functional groups (Figure 3c). The region between 3600 and 3200 cm^−1^ is typically associated with a stretching vibration of the O-H group, which leads to the formation of hydrogen bonds [22]. With the molecular weight increasing, the broadening of absorption peaks between 3600 and 3200 cm^−1^ is attributed to the presence of O-H groups and the formation of multiple hydrogen bonds amongst the O-H groups and solvent molecules [44]. It is therefore hypothesized that the 30–50 kDa and >50 kDa fractions contain more polyphenols and other hydroxyl-containing compounds, which enhance intermolecular interactions. The absorption peak at 2930 cm^−1^ corresponds to the C-H stretching vibration of the methylene group, while the peak at 1600 cm^−1^ represents the stretching vibration of the C=C bond, likely due to aromatic or conjugated systems present in the melanoidins, that contained furan, pyrrole and pyrrolinone structures and derivatives [25]. The weak peak at 1230 cm^−1^ is usually regarded as the C-N stretching vibration and N-H deformation vibration of the amide III band (1300–1200 cm^−1^) [44]. Notably, the 1–3 kDa fraction shows no absorption peaks in this range, while the peaks gradually enhance with increasing molecular weight, suggesting that higher molecular weight melanoidins contain more amide compounds. The absorption peaks in the 1000–1200 cm^−1^ range are associated with C-O stretching vibrations, particularly from the ether (C-O-C) and ester (C-O-C=O) bonds. For instance, in polysaccharides, proteins, or certain polymers, C-O bond stretching can result in strong absorption peaks [45]. The strong absorption peak at 1040 cm^−1^ in the >50 kDa fraction suggests an increased presence of polysaccharides and sugar esters, leading to a higher number of ether bonds (C-O-C). Additionally, the stretching vibrations of C-S and O-N=O groups can generate absorption peaks of around 610 cm^−1^ [45]. In summary, soy sauce melanoidins contain a variety of functional groups, such as hydroxyl (O-H), amino (N-H), aromatic ring (C=C), and sugar (C-O). These structural units contribute to the antioxidant activities of melanoidins by inhibiting the generation of free radicals during oxidation. This is consistent with the findings regarding vinegar melanoidins [30].

#### 3.5.4. 1H NMR Spectral Analysis

*1H NMR* spectra of the five melanoidins fractions exhibit a complex set of signals (Figure 3d). As the molecular weight increases, the signal peaks of melanoidins become progressively broader. This broadening is attributed to the fact that higher molecular weight prolongs the relaxation time of nuclear spins, with larger molecules—such as proteins or polymers—having longer relaxation times due to their reduced mobility [46]. As demonstrated in coffee melanoidins, the *1H NMR* spectra of melanoidins can be divided into two main regions: aliphatic hydrogen atoms (0.5–5.5 ppm) and olefinic/aromatic hydrogen atoms (6.0–8.0 ppm) [46]. The signal peaks in the 0.5–2.5 ppm range are primarily attributed to aliphatic hydrogens in proteins. As shown in Figure 3d, the >50 kDa melanoidins fraction shows the highest peaks in this region, suggesting the melanoidins fraction has a higher protein content. Additionally, the presence of amino acids, sugars, phenols and other low molecular fragments may contribute to higher fluorescence signal intensities. In the 3.0–4.5 ppm region, the highly overlapping resonance signals, due to the shielding effect of the hydroxyl group, are assigned to CH-O and CH_2_-O groups [47]. The resonance signals in the 3.5–4.5 ppm range suggest that soy sauce melanoidins contain melanoidins with intact sugar skeletons. Furthermore, the signal range of 4.5–5.0 ppm is attributed to the O-H groups in phenolic compounds [29]. Overall, the *1*H NMR spectra provide a comprehensive structural characterization of soy sauce melanoidins, and the findings are consistent with FTIR analysis.

### 3.6. Relationship Between the Structures and Antioxidant Activities of Melanoidins

Spectral analysis reveals that the structural characteristics of melanoidins fractions with different molecular weights directly influence their antioxidant activities. As shown in Table 5, low molecular weight fractions (1–3 kDa and 3–10 kDa) are primarily composed of amino acids and peptides, with fewer conjugated systems and active functional groups. The 1–3 kDa and 3–10 kDa fractions, rich in amino-terminal peptide segments and partially exposed phenolic hydroxyl groups, exhibit excellent free radical scavenging capacity due to their small molecular size, low steric hindrance, and high diffusion coefficient. However, their performance in MCA and FRAP assays is relatively weak due to the limited availability of hydroxyl (O-H) groups and conjugated systems. Additionally, nitrogen atoms in the nitrogen-containing structures of low molecular weight melanoidins can undergo cross-linking polymerization with furan subunits derived from carbohydrates through amino and guanidino groups of amino acids, resulting in the formation of complex high molecular weight fractions [42,48,49]. This phenomenon has also been observed in bread, coffee, and other Maillard reaction products [10,14].

High molecular weight melanoidins fractions (10–30 kDa, 30–50 kDa, >50 kDa) contain abundant conjugated double bonds and aromatic systems (e.g., furan/pyrrole rings), which stabilize free radical intermediates through π-π conjugation-mediated electron delocalization [36,44]. This electron delocalization is the structural basis for their antioxidant function, facilitating multi-level inhibition of oxidative stress by stabilizing free radical intermediates, enhancing electron transfer efficiency, and synergizing with other functional groups [38,50]. Specifically, the 10–30 kDa fraction, rich in aromatic rings, C=C/C=O, and O-H groups, significantly promotes Fe^3+^ reduction via π-π conjugation and O-H-mediated electron transfer. The 30–50 kDa fraction exhibits strong Fe^2+^/Cu^2+^ chelation capacity and high FRAP activity through the synergistic interaction of C=C, amide (N-H), and O-H groups. Although the >50 kDa fraction is spatially hindered by its complex structure, its abundance of active groups, polysaccharides, and protein-based macromolecular aggregation structures enables it to maintain strong metal chelation and reduction capabilities.

In conclusion, the internal structural aggregation levels of melanoidins fractions with different molecular weights and the differences in the content of conjugated systems (C=C, C=O) and active functional groups (N-H, O-H) directly contribute to the differences in antioxidant activity across the fractions. However, the synergistic interactions between the structures of melanoidins with different molecular weight fractions enhance the antioxidant functionality of soy sauce melanoidins and play a key role in contributing to the antioxidant activity of soy sauce [49].

## 4. Conclusions

This study systematically investigated the antioxidant activities and structure characteristics of soy sauce melanoidins. The results indicated that soy sauce melanoidins primarily consisted of proteins, sugars, and phenolic compounds, with proteins being the predominant component. The five fractions of soy sauce melanoidins had significant differences in antioxidant contributions to soy sauce, demonstrating that 30–50 kDa and >50 kDa melanoidins fractions were the key contributors to the overall antioxidant activities of soy sauce. The conjugated structures and reactive groups, i.e., C=C, C=O, N-H, O-H, in melanoidins were crucial for the antioxidant activities of soy sauce. This study not only deepened our understanding of the antioxidant potential and mechanisms of soy sauce melanoidins but also provides theoretical support for the regulation of antioxidant active components and the development of high-value-added functional seasonings in the deep processing of fermented food. Currently, research on the synergistic effect of melanoidins with different molecular weights and other antioxidant components on the antioxidant activities of soy sauce is ongoing.

## Figures and Tables

**Figure 1 foods-14-02787-f001:**
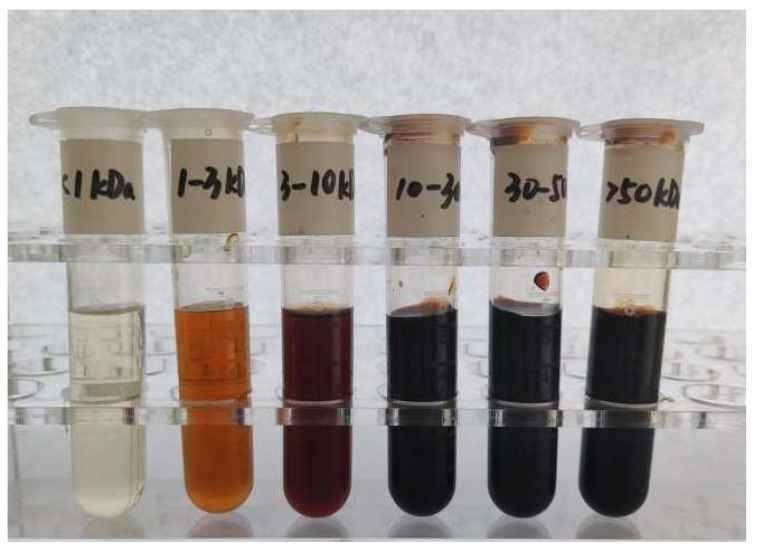
Melanoidins extracted by resin and separated by ultrafiltration.

**Figure 2 foods-14-02787-f002:**
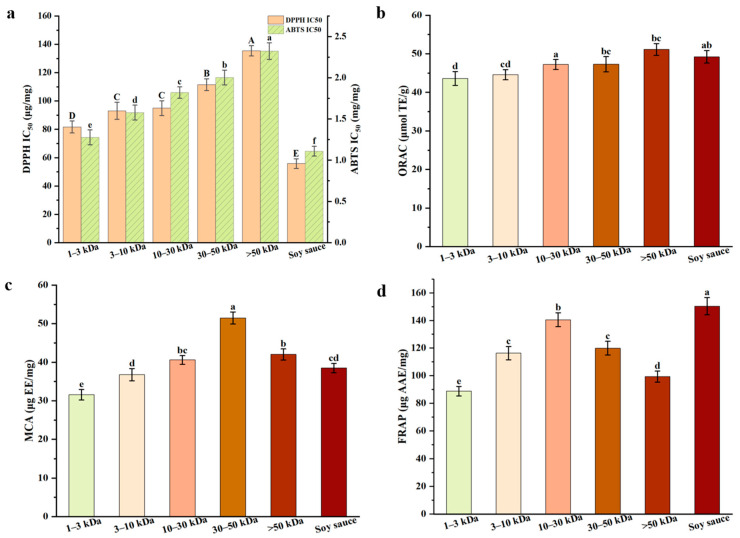
Antioxidant activities of melanoidins fractions with different molecular weights: (**a**) DPPH and ABTS radical scavenging activity; (**b**) oxygen radical absorbing capacity; (**c**) metal ion chelating activity; (**d**) ferric reducing antioxidant power. ^a–f (A–D)^ Different letters in the same row indicate the significant differences (*p* < 0.05).

**Figure 3 foods-14-02787-f003:**
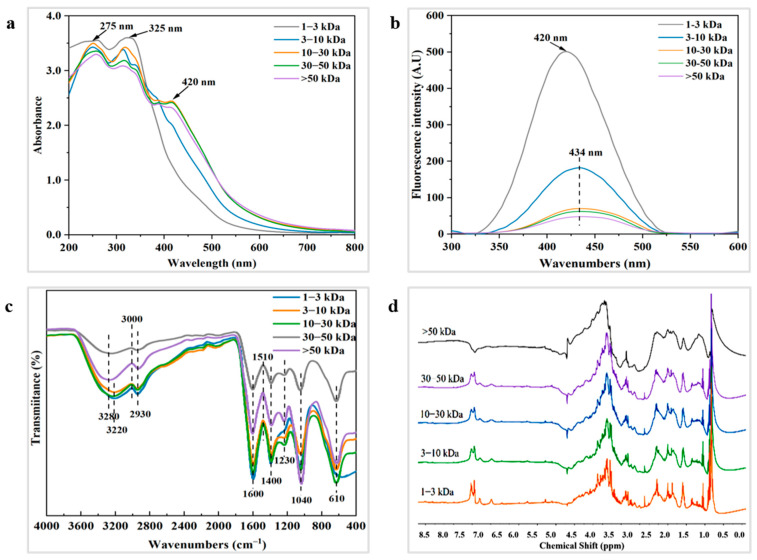
Structural characteristics of melanoidins with different molecular weights: (**a**) UV–vis-spectra; (**b**) fluorescence spectra; (**c**) FTIR spectra; (**d**) 1*H NMR* spectra.

**Table 1 foods-14-02787-t001:** Content of melanoidins fractions with different molecular weights in soy sauce.

Melanoidins	Molecular Weight Fraction				
	1–3 kDa	3–10 kDa	10–30 kDa	30–50 kDa	>50 kDa
Content (g/100 mL soy sauce)	0.31 ± 0.07 ^c,d^	0.42 ± 0.09 ^b,c^	0.20 ± 0.05 ^d^	0.61 ± 0.10 ^b^	0.90 ± 0.13 ^a^

^a–d^ Different letters in the same row indicate the significant differences (*p* < 0.05).

**Table 2 foods-14-02787-t002:** Main components of melanoidins fractions with different molecular weights.

Main Components	Molecular Weight Fraction				
	1–3 kDa	3–10 kDa	10–30 kDa	30–50 kDa	>50 kDa
Total sugar (mg/g)	16.03 ± 0.61 ^c^	17.06 ± 1.03 ^c^	17.34 ± 1.11 ^c^	22.18 ± 1.63 ^b^	28.34 ± 1.90 ^a^
Protein (mg/g)	50.18 ± 3.12 ^d^	50.81 ± 2.07 ^d^	71.83 ± 4.03 ^c^	83.37 ± 4.15 ^b^	109.20 ± 7.22 ^a^
Total phenol (mg/g)	0.20 ± 0.05 ^c^	0.56 ± 0.01 ^b^	0.88 ± 0.12 ^a^	0.55 ± 0.08 ^b^	0.43 ± 0.06 ^b^

^a–d^ Different letters in the same row indicate the significant differences (*p* < 0.05).

**Table 3 foods-14-02787-t003:** Free amino acids of melanoidins fractions with different molecular weights.

Free Amino Acids (g/100 g)	Molecular Weight Fraction				
	1–3 kDa	3–10 kDa	10–30 kDa	30–50 kDa	>50 kDa
Aspartic acid	0.44 ± 0.03 ^a^	0.45 ± 0.04 ^a^	0.46 ± 0.02 ^a^	0.46 ± 0.05 ^a^	0.37 ± 0.02 ^b^
Glutamic acid	1.82 ± 0.17 ^a^	1.47 ± 0.07 ^b^	1.09 ± 0.18 ^c^	0.96 ± 0.06 ^c^	0.56 ± 0.04 ^d^
Serine	0.09 ± 0.01 ^b^	0.09 ± 0.01 ^b,c^	0.06 ± 0.01 ^d^	0.07 ± 0.01 ^c,d^	0.17 ± 0.01 ^a^
Glycine	0.12 ± 0.01 ^a^	0.08 ± 0.01 ^b^	0.07 ± 0.01 ^b^	0.07 ± 0.01 ^b^	0.07 ± 0.01 ^b^
Threonine	0.08 ± 0.01 ^a^	0.06 ± 0.01 ^b^	0.06 ± 0.01 ^b^	0.05 ± 0.01 ^b^	0.06 ± 0.01 ^b^
Arginine	0.44 ± 0.03 ^a^	0.37 ± 0.02 ^b^	0.28 ± 0.02 ^c^	0.31 ± 0.02 ^c^	0.20 ± 0.40 ^d^
Alanine	0.07 ± 0.01 ^a^	0.04 ± 0.01 ^b^	0.05 ± 0.01 ^a,b^	0.03 ± 0.01 ^b,c^	0.02 ± 0.01 ^c^
Tyrosine	0.30 ± 0.02 ^a^	0.17 ± 0.01 ^b^	0.11 ± 0.01 ^c^	0.13 ± 0.01 ^c^	0.06 ± 0.02 ^d^
Cysteine	0.02 ± 0.01 ^a^	0.01 ± 0.01 ^a^	0.01 ± 0.01 ^a^	0.01 ± 0.01 ^a^	0.01 ± 0.01 ^a^
Valine	0.37 ± 0.02 ^a^	0.23 ± 0.02 ^b^	0.15 ± 0.01 ^c^	0.17 ± 0.01 ^c^	0.06 ± 0.02 ^d^
Isoleucine	1.07 ± 0.08 ^a^	0.76 ± 0.05 ^b^	0.55 ± 0.03 ^c^	0.47 ± 0.03 ^c^	0.21 ± 0.08 ^d^
Leucine	1.62 ± 0.15 ^a^	1.27 ± 0.09 ^b^	0.93 ± 0.07 ^c^	0.78 ± 0.06 ^c^	0.35 ± 0.15 ^d^
Methionine	0.17 ± 0.01 ^a^	0.11 ± 0.01 ^b^	0.07 ± 0.01 ^c^	0.08 ± 0.01 ^c^	0.03 ± 0.01 ^d^
Lysine	0.20 ± 0.01 ^a^	0.17 ± 0.01 ^b^	0.21 ± 0.01 ^a^	0.15 ± 0.01 ^c^	0.09 ± 0.01 ^d^
Proline	0.11 ± 0.01 ^a^	0.07 ± 0.01 ^b^	0.07 ± 0.01 ^b^	0.07 ± 0.01 ^b^	0.04 ± 0.01 ^c^
Tryptophan	0.25 ± 0.02 ^a^	0.13 ± 0.01 ^b^	0.08 ± 0.01 ^c^	0.07 ± 0.01 ^c,d^	0.05 ± 0.01 ^d^
Total	8.92	6.82	5.27	4.75	2.93

^a–d^ Different letters in the same row indicate the significant differences (*p* < 0.05).

**Table 4 foods-14-02787-t004:** Contribution of melanoidins fractions with different molecular weights to antioxidant activities of soy sauce.

Antioxidant Assays	Melanoidins Molecular Weight Fraction Solution					Soy Sauce
	1–3 kDa	3–10 kDa	10–30 kDa	30–50 kDa	>50 kDa	
DPPH (μg AAE/mL)	0.81 ± 0.04 ^e^	1.47 ± 0.05 ^c^	0.91 ± 0.06 ^e^	1.17 ± 0.03 ^d^	1.80 ± 0.05 ^b^	9.25 ± 0.26 ^a^
ABTS (μmol TE/mL)	0.04 ± 0.01 ^c^	0.05 ± 0.01 ^b,c^	0.03 ± 0.01 ^c^	0.05 ± 0.01 ^b,c^	0.07 ± 0.01 ^b^	0.52 ± 0.03 ^a^
OARC (μmol TE/mL)	0.71 ± 0.06 ^d^	0.89 ± 0.08 ^c,d^	0.77 ± 0.09 ^d^	1.13 ± 0.08 ^b,c^	1.38 ± 0.09 ^b^	8.34 ± 0.28 ^a^
MCA (μg EE/mL)	0.78 ± 0.09 ^d^	2.08 ± 0.14 ^c^	0.96 ± 0.10 ^d^	2.30 ± 0.17 ^c^	3.17 ± 0.21 ^b^	12.38 ± 0.35 ^a^
FRAP (μg AAE/mL)	0.92 ± 0.04 ^e^	2.94 ± 0.16 ^c^	1.51 ± 0.05 ^d^	3.37 ± 0.17 ^c^	4.42 ± 0.21 ^b^	38.46 ± 1.15 ^a^
Contribution to DPPH in soy sauce (%)	8.56 ± 0.46 ^e^	15.89 ± 0.11 ^b^	9.84 ± 0.06 ^d^	12.64 ± 0.13 ^c^	19.46 ± 0.28 ^a^	—
Contribution to ABTS in soy sauce (%)	7.69 ± 0.37 ^c^	9.62 ± 0.41 ^b^	5.77 ± 0.26 ^d^	9.62 ± 0.35 ^b^	13.46 ± 0.30 ^a^	—
Contribution to ORAC in soy sauce (%)	8.51 ± 0.53 ^d^	10.67 ± 0.50 ^c^	9.23 ± 0.49 ^d^	13.55 ± 0.62 ^b^	16.55 ± 0.68 ^a^	—
Contribution to MCA in soy sauce (%)	6.30 ± 0.26 ^e^	16.80 ± 0.49 ^c^	7.74 ± 0.42 ^d^	18.58 ± 0.51 ^b^	25.61 ± 0.45 ^a^	—
Contribution to FRAP in soy sauce (%)	2.39 ± 0.11 ^e^	7.64 ± 0.14 ^c^	3.93 ± 0.09 ^d^	8.76 ± 0.08 ^b^	11.49 ± 0.12 ^a^	—

The antioxidant activity (*M_i_*) is first normalized per milliliter of soy sauce (μg AAE, μmol TE, or μg EE) and then converted to percentage contributions (*W_i_*) using Equation (4). Values are means ± SD (*n* = 3); ^a–e^ Different letters in the same row indicate the significant differences (*p* < 0.05).

**Table 5 foods-14-02787-t005:** The main functional groups of melanoidins fractions with different molecular weights.

Melanoidins Molecular Weight Fraction	Main Functional Groups
1–3 kDa	Amino acids, small peptides, N-H, no clear C=C or C=O peaks
3–10 kDa	Amino acids, small peptides, N-H, some C=C, C=O groups
10–30 kDa	C=C, C=O, O-H, N-H, polyphenols, aromatic rings
30–50 kDa	C=C, C=O, O-H, N-H, polyphenols, aromatic rings, amides, polysaccharides
>50 kDa	C=C, C=O, O-H, N-H, polysaccharides, polyphenols, amides, aromatic rings, esters

## Data Availability

The original contributions presented in this study are included in the article. Further inquiries can be directed to the corresponding author.

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
