# Peer review of "The Contribution of Melanoidins to Soy Sauce Antioxidant Activities and Their Structure Characteristics"

_foods, 2025, doi:10.3390/foods14162787_

Round 1

Reviewer 1 Report

Comments and Suggestions for Authors

This manuscript reports the fractionation of melanoidins from soy sauce and the evaluation of their antioxidant activity using chemical methods. In addition, several spectroscopic analyses were performed to identify some functional groups and molecular arrangements related to melanoidin structures. The study provides insights into the antioxidant properties of soy sauce and its melanoidin fractions; however, the results and discussion should be improved to better demonstrate their relevance and accuracy

Introduction

Line 31-32: use italics letter for Aspergillus oryzae name

Lines 33-36: Temperature plays a key role for the Maillard, in this context it is relevant to cite the temperature range using during soy sauce production

Line 55: Please clarify that this per capital intake refers to Asian population

Line 64: type mistake “fermentation.4”

Methods

Line 86: type mistake “Gao et al.,2”, please to cite the correspond reference

Lines 93-94: Please specify how many milliliters of ethanol were passed through the resin column.

Section 2.3: Why was a standard curve not used for this purpose?

Results – Discussion

Lines 273-277: Could the author please explain these results? I believe these findings are related to the molecular weight of the sugars and proteins/peptides, which were also fractionated using the separation resin

Lines 327-330: I do not agree with this explanation. If it were correct, similar results would be expected in the FRAP assay.

Lines 343-356: What is the relationship between the high phenolic content found in the 10–30 kDa and 30–50 kDa fractions and the results observed in the MCA and FRAP assays?

Section 3.4: These results do not agree with those presented in Section 3.3; they appear confusing and contradictory. Please include a suitable explanation for these findings or consider removing them from the manuscript

Section 3.5:

  • I suggest including a table in the manuscript that summarizes the functional groups or molecular structures identified based on the spectroscopy results.
  • According to the spectroscopy results, what could be the main differences between the melanoidins present in each fraction? Please include an explanation in the manuscript

Lines 505-508: I consider that the results reported in Section 3.5 are not sufficient or adequate to support this conclusion. In its current form, Section 3.6 is merely a theoretical discussion without sufficient experimental support. Therefore, it is recommended that this section be removed from the manuscript, unless additional experimental evidence can be provided to support the discussion

Author Response

Response to the editor and reviewers

Dear Reviewers,

We are thankful to the editor and the reviewers for your valuable suggestions on your manuscript, which helped us to improve the quality of our manuscript (foods-3748081). We have carefully revised the manuscript according to the suggestions point by point and addressed all the concerns. All the changes were highlighted in the revised manuscript.

Thank you very much for your hard work on our manuscript.

Sincerely yours,

Xianli Gao

Reviewer 1

Comments:

This manuscript reports the fractionation of melanoidins from soy sauce and the evaluation of their antioxidant activity using chemical methods. In addition, several spectroscopic analyses were performed to identify some functional groups and molecular arrangements related to melanoidin structures. The study provides insights into the antioxidant properties of soy sauce and its melanoidin fractions; however, the results and discussion should be improved to better demonstrate their relevance and accuracy.

Answer (A): Thank you for your valuable feedback on this manuscript. We fully agree with your assessment and have made comprehensive revisions to the results and discussion sections to improve their relevance and accuracy.

To more clearly highlight the relevance of our findings, we have provided a detailed explanation of the structural characteristics of the different molecular weight melanoidins fractions identified through spectral analysis and their association with antioxidant activities. Additionally, we have elucidated the mechanistic relationship between functional groups, such as conjugated double bonds and hydroxyl groups, and antioxidant activity, thereby strengthening the theoretical framework. In accordance with your detailed suggestions, we have completed the revisions. The modified sections are highlighted in red in the manuscript.

Introduction

Line 31-32: use italics letter for Aspergillus oryzae name

A: Thanks for your kind suggestion. We have revised the manuscript and standardized the use of italics for all microbial names mentioned throughout the manuscript.

Lines 33-36: Temperature plays a key role for the Maillard, in this context it is relevant to cite the temperature range using during soy sauce production

A: Thanks for your kind suggestion. The fermentation of soy sauce is conducted outdoor and the fermentation temperature of moromi ranges from 20 °C to 40 °C from spring to autumn. We have therefore revised the sentence as follows:

'In the subsequent moromi fermentation, the mash is placed outdoors to ferment in an 18–20% (w/v) brine for six months at the temperatures that fluctuate between 20 °C and 40 °C. The condition promotes the Maillard reaction and other biochemical transformations that contribute to the characteristic flavor and color of soy sauce [2-4].'

Line 55: Please clarify that this per capital intake refers to Asian population

A: Thanks for your kind suggestion. We have revised the sentence as follows:

'Notably, soy sauce, a common edible condiment with a per capita intake of 10–20 g per day, particularly among Asian populations, exhibits significantly higher free radical scavenging capacity than several other fermented products.'

Line 64: type mistake “fermentation.4”

A: Thanks for your kind suggestion. We have corrected the reference citation as follows: ' fermentation. [4]'

Methods

Line 86: type mistake “Gao et al.,2”, please to cite the correspond reference

A: Thanks for your kind suggestion. We have corrected the citation as follows: 'Gao et al. [2].' The reference has been updated accordingly in the manuscript.

Lines 93-94: Please specify how many milliliters of ethanol were passed through the resin column.

A: Thanks for your kind suggestion. The revised manuscript now specifies that 200 mL of 40 % ethanol (equivalent to 4 bed volumes) was used for elution. The revised sentence is as the following:

'Once adsorption equilibrium was achieved, the resin was eluted with 200 mL of 40% (v/v) ethanol (equivalent to 4 bed volumes) at a flow rate of 3 mL/min.'

Section 2.3: Why was a standard curve not used for this purpose?

A: Thanks for your kind suggestion. Due to the complex and heterogeneous nature of melanin, no standard samples are available on the market, and, therefore, no standard curve has been established. We referenced the quantitative determination methods for melanoidins proposed by Mossine et al. [23] and Li et al. [24], and defined the molar extinction coefficient (0.269 AU, corresponding to 0.1 mg/mL) using a model melanoidins solution. This coefficient has been validated in beer and vinegar melanoidins [12,38], which allows for the estimation of melanoidins content.

Results – Discussion

Lines 273-277: Could the author please explain these results? I believe these findings are related to the molecular weight of the sugars and proteins/peptides, which were also fractionated using the separation resin.

A: Thank you for your valuable suggestion. We agree with your perspective that sugars, proteins, and peptide segments are separated based on the principle of fractionation. It is essential to note that prior to ultrafiltration separation, melanoidins in soy sauce were purified and separated using macroporous resin adsorption. This process removed most free sugars and proteins that did not participate in the Maillard reaction, while retaining the macromolecular substances that form covalent or non-covalent cross-links with melanoidins. Therefore, the high molecular weight sugars and proteins detected in the various melanoidin components (Table 2) are not free impurities but rather integral components of the melanoidin backbone. These components cross-link with low molecular weight units during the Maillard reaction to form the final melanoidins polymer, with the highest content observed in the high molecular weight fraction (>50 kDa), which is consistent with our research result. Since these sugar and protein/peptide complexes are incorporated into the melanoidins structure, ultrafiltration separation does not dilute or enrich any free components. Therefore, the melanoidins content in each molecular weight fraction accurately reflects its true distribution.

Lines 327-330: I do not agree with this explanation. If it were correct, similar results would be expected in the FRAP assay.

A: Thank you for your valuable suggestion. We appreciate the opportunity to clarify the apparent discrepancy between DPPH/ABTS and FRAP assay results, which arises from the fundamental differences in their antioxidant mechanisms.

The key distinction lies in the fact that DPPH and ABTS assays primarily assess hydrogen atom transfer (HAT) reactions. In these assays, low molecular weight fractions, such as small peptides and free amino acids, can readily collide with free radicals and efficiently donate hydrogen atoms due to their smaller molecular weight, lower steric hindrance, and higher diffusion coefficient. In contrast, the FRAP assay measures electron transfer (ET) capacity, which depends on conjugated systems, such as C=C bonds and aromatic rings, that facilitate electron delocalization. High molecular weight fractions (10–30 kDa, 30–50 kDa, >50 kDa) are rich in π-π conjugated double bonds, aromatic rings, and hydroxyl groups, driving efficient electron transfer and enhancing the ET-mediated reduction of Fe³⁺ to Fe²⁺. This is consistent with previous studies, which show that electron transfer-dependent assays (e.g., FRAP) are strongly correlated with conjugated structures in high molecular weight melanoidins, while HAT-dependent assays (DPPH/ABTS) favor small-molecule, highly mobile antioxidants. To avoid ambiguity, we have revised some of the statements in the manuscript:

'3.3 Antioxidant activities of melanoidins

'... and it is hypothesized that this discrepancy may stem from differences between the chemical structure of melanoidins and molecular weight. Unlike high-temperature short-term fermentation, low-temperature long-term fermentation of soy sauce substantially slows the formation of Maillard re-action products, promoting the accumulation of low molecular weight components such as small peptides and free amino acids [6,8]. Due to their smaller molecular size, lower steric hindrance, and higher diffusion coefficient, they have an increased collision frequency with free radicals. They effectively scavenge DPPH/ABTS free radicals through the hydrogen atom transfer (HAT) mechanism, resulting in lower IC₅₀ values…

…. Moreover, as shown in Figure. 3d, the FRAP assay measures electron transfer (ET) capacity, which depends on the con-jugated system to promote electron delocalization [28]. High molecular weight fractions (10–30 kDa, 30–50 kDa, >50 kDa) are rich in π-π conjugated double bonds, aromatic rings, and hydroxyl groups, driving efficient electron transfer and enhancing Fe³⁺ reduction to Fe²⁺ mediated by ET. Among these, the 10–30 kDa fraction exhibits the highest FRAP value (140.59 μg AAE/mg), which is significantly higher than that of the 1–3 kDa fraction (88.88 μg AAE/mg) and the 3–10 kDa fraction (116.39 μg AAE/mg) (p < 0.05) …'

Lines 343-356: What is the relationship between the high phenolic content found in the 10–30 kDa and 30–50 kDa fractions and the results observed in the MCA and FRAP assays?

A: Thanks for your kind suggestion. The high phenolic content in the 10–30 kDa and 30–50 kDa fractions enhances their metal ion chelation and reducing capacity. Phenolic compounds, such as those found in melanoidins, contain hydroxyl and ketone groups that serve as chelation donors. These functional groups are essential for binding metal ions, particularly Fe²⁺, and for facilitating the reduction of Fe³⁺ to Fe²⁺. The strong chelation ability of the high molecular weight fractions, in conjunction with their higher phenolic content, results in more effective metal ion binding and reduced oxidative stress. This explains why these fractions exhibit significantly higher MCA and FRAP values compared to lower molecular weight fractions, such as the 1–3 kDa fraction.

Section 3.4: These results do not agree with those presented in Section 3.3; they appear confusing and contradictory. Please include a suitable explanation for these findings or consider removing them from the manuscript

A: Thanks for your kind suggestion. We apologize for the confusion caused by the initial presentation of Sections 3.3 and 3.4. These two sections address different aspects of melanoidins' antioxidant capacities, and their results are complementary rather than contradictory. Section 3.3 focuses on evaluating the antioxidant activities of different fractions at the same concentration. The objective is to compare their inherent antioxidant capacity without being influenced by their content in soy sauce. In contrast, Section 3.4 evaluates the actual contribution of each fraction to the total antioxidant activity of soy sauce based on the actual concentration (Table 1) of each fraction in soy sauce. Additionally, we have made the following clarification at the beginning of Section 3.4:

'3.4 Contributions of melanoidins to the antioxidant activities of soy sauce

Based on the actual concentrations of five fractions in soy sauce (Table 1), corresponding melanoidins fraction solutions were prepared to assess the contribution of each fraction to the overall antioxidant activity of soy sauce. The results of the antioxidant assays, conducted using five antioxidant assays (DPPH, ABTS, ORAC, MCA, and FRAP), are presented in Table 4….'

Section 3.5:

I suggest including a table in the manuscript that summarizes the functional groups or molecular structures identified based on the spectroscopy results.

A: Thanks for your kind suggestion. We have added Table 5 in Section 3.6, summarizing the main functional groups identified from the spectroscopic results of different molecular weight melanoidins fractions. able 5 is as the following:

Table 5. The main functional groups of melanoidins fractions with different molecular weights.

Melanoidins Molecular Weight Fraction

Main Functional Groups

1–3 kDa

Amino acids, small peptides, N-H, no clear C=C or C=O peaks

3–10 kDa

Amino acids, small peptides, N-H, some C=C, C=O groups

10–30 kDa

C=C, C=O, O-H, N-H, polyphenols, aromatic rings

30–50 kDa

C=C, C=O, O-H, N-H, polyphenols, aromatic rings, amides, polysaccharides

>50 kDa

C=C, C=O, O-H, N-H, polysaccharides, polyphenols, amides

According to the spectroscopy results, what could be the main differences between the melanoidins present in each fraction? Please include an explanation in the manuscript

A: Thanks for your kind suggestion. We have provided a detailed explanation of the main differences between the five fractions of melanoidins in the manuscript, integrating Table 5 and Section 3.6. The low molecular weight fractions (1–3 kDa, 3–10 kDa) primarily consist of amino acids and small peptides, with few conjugated systems. In contrast, the high molecular weight fractions (10–30 kDa, 30–50 kDa, >50 kDa) contain a greater number of conjugated systems (C=C, C=O) and active functional groups (N-H, O-H), exhibiting a higher degree of polymerization and more complex structures. As the molecular weight increases, these structural differences directly influence the varying antioxidant activities.

Lines 505-508: I consider that the results reported in Section 3.5 are not sufficient or adequate to support this conclusion. In its current form, Section 3.6 is merely a theoretical discussion without sufficient experimental support. Therefore, it is recommended that this section be removed from the manuscript, unless additional experimental evidence can be provided to support the discussion

A: Thanks for your kind suggestion. We acknowledge your concern regarding the excessive reliance on theoretical frameworks in Section 3.6. In response, we have revised this section to eliminate purely theoretical discussions. Instead, we focus on examining the relationship between the structural characteristics of melanoidins fractions with different molecular weights, and their antioxidant activity, in light of current experimental results.

'3.6 Relationship between the structures and antioxidant activities of melanoidins

Spectral analysis reveals that the structural characteristics of melanoidins fractions with different molecular weights directly influence their antioxidant activities. As shown in Table 5, low molecular weight fractions (1–3 kDa and 3–10 kDa) are primarily composed of amino acids and peptides, with fewer conjugated systems and active functional groups. The 1–3 kDa and 3–10 kDa fractions, rich in amino-terminal peptide segments and partially exposed phenolic hydroxyl groups, exhibit excellent free radical scavenging capacity due to their small molecular size, low steric hindrance, and high diffusion coefficient. However, their performance in MCA and FRAP assays is relatively weak due to the limited availability of hydroxyl (O-H) groups and conjugated systems. Additionally, nitrogen atoms in the nitrogen-containing structures of low molecular weight melanoidins can undergo cross-linking polymerization with furan subunits derived from carbohydrates through amino and guanidino groups of amino acids, resulting in the formation of complex high molecular weight fractions [42,48-49]. This phenomenon has also been observed in bread, coffee, and other Maillard reaction products [10,14].

High molecular weight melanoidins fractions (10–30 kDa, 30–50 kDa, >50 kDa) contain abundant conjugated double bonds and aromatic systems (e.g., furan/pyrrole rings), which stabilize free radical intermediates through π-π conjugation-mediated electron delocalization [36,44]. This electron delocalization is the structural basis for their antioxidant function, facilitating multi-level inhibition of oxidative stress by stabilizing free radical intermediates, enhancing electron transfer efficiency, and synergizing with other functional groups [38,50]. Specifically, the 10–30 kDa fraction, rich in aromatic rings, C=C/C=O, and O-H groups, significantly promotes Fe³⁺ reduction via π-π conjugation and O-H-mediated electron transfer. The 30–50 kDa fraction exhibits strong Fe²⁺/Cu²⁺ chelation capacity and high FRAP activity through the synergistic interaction of C=C, amide (N-H), and O-H groups. Although the >50 kDa fraction is spatially hindered by its complex structure, its abundance of active groups, polysaccharides, and protein-based macromolecular aggregation structures enables it to maintain strong metal chelation and reduction capabilities.

In conclusion, the internal structural aggregation levels of melanoidins fractions with different molecular weights, and there are differences in the content of conjugated systems (C=C, C=O) and active functional groups (N-H, O-H), which directly con-tribute to the differences in antioxidant activity across the fractions. However, the synergistic interactions between the structures of melanoidins with different molecular weight fractions enhance the antioxidant functionality of soy sauce melanoidins and plays a key role in contributing to the antioxidant activity of soy sauce [49].'

Reviewer 2

Comments:

Detailed Points for Improvement

Keywords: Avoid using keywords that already appear in the title of the manuscript.

A: Thanks for your kind suggestion. We have revised the keywords to avoid repetition with the title in the manuscript. The revised keywords now read:

'Keywords: Maillard reaction; soy sauce melanoidins; molecular weight; bioactivity; antioxidant contributions; functional food'

Materials and methods / Discussion:

Deepen the discussion of differences between soy sauce melanoidins and those from other sources (e.g., coffee, bread) beyond what is already mentioned.

A: Thanks for your kind suggestion. The main difference between the Maillard reaction products of soy sauce and those from other sources (such as coffee and bread) lies in their chemical composition. We have provided more details in Section 3.2.1:

'3.2.1 The main composition of melanoidins

…this is due to the fact that soy sauce is typically fermented from protein-rich raw materials such as soybeans and wheat, which are enzymatically hydrolyzed to produce amino acids and small peptides, serving as precursors for the formation of soy sauce melanoidins [3,32]. This is the reason for the higher proportion of protein in soy sauce melanoidins compared with other foods like coffee and chocolate [13,33].

… Compared to the content of phenolic in coffee, chocolate, and sweet wine [33-34], the phenolics content of soy sauce melanoidins is lower. This is primarily attributed to the fact that grain-derived fermented foods have lower phenolic content, resulting in melanoidins proteins with reduced cross-reactivity between gluten, sugars, and phenolic substance…'

Address the practical implications of the findings more explicitly.

A: Thanks for your kind suggestion. I have refined the practical implications of the research findings in the revised draft and summarized them in the conclusion section:

'4. Conclusions

… This study not only deepened our understanding of the antioxidant potential and mechanisms of soy sauce melanoidins but also provides theoretical support for the regulation of antioxidant active components and the development of high-value-added functional seasonings in the deep processing of fermented food…'

In the Materials and Methods section, the authors state that the results for the DPPH and ABTS assays were expressed as ascorbic acid equivalents (AAE) for DPPH (µg AAE/mL) and Trolox equivalents (TE) for ABTS (µmol TE/mL). However, in the Results and Discussion section, these values are later presented as percentages of contribution to the antioxidant activity of soy sauce, without a clear connection between the initially reported units and this percentage representation. Moreover, there is a lack of clarity on how the authors transitioned from the absolute values (expressed in AAE or TE) to the relative percentage contributions for each molecular weight fraction. This inconsistency between the description of the methods and the presentation of the results creates confusion about how the data were processed and compared. A clearer explanation of the conversion process, including any formulas or normalization steps used to derive these percentages, would improve the transparency and reproducibility of the findings. Additionally, the authors refer to IC50 values (concentration required to scavenge 50% of free radicals) in the Results and Discussion section, particularly when describing the antioxidant activity of the different melanoidin fractions in the DPPH and ABTS assays. However, there is no mention of IC50 determination or any corresponding methodology in the Materials and Methods section. The methods only describe how antioxidant activities were expressed in terms of ascorbic acid equivalents (AAE) or Trolox equivalents (TE), but do not explain how the IC50 values were calculated, what software or statistical tools were used, or how the dose-response curves were constructed. Therefore, clarification is needed. The authors should explicitly describe in the Methods how IC50 values were determined if they intend to present and discuss these results. Otherwise, including IC50 values in the Results without prior methodological description is inconsistent and undermines the transparency of the study.

A: Thanks for your kind suggestion. Firstly, to clarify the contribution ratios of different molecular weight components to the antioxidant activity of soy sauce, we have detailed the calculation process in Section 2.6 and provided a brief explanation below Table 4. Secondly, to clarify the presentation format of the DPPH and ABTS assay results, we have provided detailed explanations regarding the calculation of IC50 values in the methods section of Sections 2.5.1 and 2.5.2.

'2.6 Contributions of melanoidins to the antioxidant activities of soy sauce

... Antioxidant activities of the melanoidins fraction solution and soy sauce were assessed using the methods described in Section 2.5 (DPPH、FRAP、ABTS 、 ORAC and MCA). The percentage contribution of each melanoidins fraction to the overall antioxidant activity of soy sauce was calculated using Equation (4):

where Mi is the antioxidant activities of melanoidins fractioni solution (i = 1–3 kDa, 3–10 kDa, 10–30 kDa, 30–50 kDa, and >50 kDa); N is the antioxidant activity of soy sauce measured under the same measurement conditions, Wi is the percentage contribution of fractioni to the antioxidant activity of soy sauce.'

Table 4. Contribution of melanoidins fractions with different molecular weights to antioxidant activities of soy sauce.

Antioxidant Assays

Melanoidins Molecular Weight Fraction Solution

soy sauce

1−3 kDa

3−10 kDa

10−30 kDa

30−50 kDa

>50 kDa

DPPH (μg AAE/mL)

0.81 ± 0.04e

1.47 ± 0.05c

0.91 ± 0.06e

1.17 ± 0.03d

1.80 ± 0.05b

9.25 ± 0.26a

ABTS (μmol TE/mL)

0.04 ± 0.01c

0.05 ± 0.01b,c

0.03± 0.01c

0.05 ± 0.01b,c

0.07 ± 0.01b

0.52 ± 0.03a

OARC (μmol TE/mL)

0.71 ± 0.06d

0.89 ± 0.08c,d

0.77 ± 0.09d

1.13 ± 0.08b,c

1.38 ± 0.09b

8.34 ± 0.28a

MCA (μg EE/mL)

0.78 ± 0.09d

2.08 ± 0.14c

0.96 ± 0.10d

2.30 ± 0.17c

3.17 ± 0.21b

12.38 ± 0.35a

FRAP (μg AAE/mL)

0.92 ± 0.04e

2.94 ± 0.16c

1.51 ± 0.05d

3.37 ± 0.17c

4.42 ± 0.21b

38.46 ± 1.15a

Contribution to DPPH in soy sauce (%)

8.56 ± 0.46e

15.89 ± 0.11b

9.84 ± 0.06d

12.64 ± 0.13c

19.46 ± 0.28a

Contribution to ABTS in soy sauce (%)

7.69 ± 0.37c

9.62 ± 0.41b

5.77 ± 0.26d

9.62 ± 0.35b

13.46 ± 0.30a

Contribution to ORAC in soy sauce (%)

8.51 ± 0.53d

10.67 ± 0.50c

9.23 ± 0.49d

13.55 ± 0.62b

16.55 ± 0.68a

Contribution to MCA in soy sauce (%)

6.30 ± 0.26e

16.80 ± 0.49c

7.74 ± 0.42d

18.58 ± 0.51b

25.61 ± 0.45a

Contribution to FRAP in soy sauce (%)

2.39 ± 0.11e

7.64 ± 0.14c

3.93 ± 0.09d

8.76 ± 0.08b

11.49 ± 0.12a

The antioxidant activities (Mᵢ) are first normalised per milliliter of soy sauce (μg AAE, μmol TE, or μg EE) and then converted to percentage contributions (Wᵢ) using Eq. (4). Values are means ± SD (n = 3). a-e Different letters in the same row indicate the significant differences (p < 0.05).

2.5.1 DPPH free radical scavenging activity

…The mixture was incubated in dark for 30 min, after which the absorbance was measured at 517 nm. A0, A1 and AS are the absorbances of the blank (sample solution was replaced with distilled water), sample and control solutions (DPPH solution was replaced with ethanol), respectively. Ascorbic acid was used as the positive control. DPPH scavenging activity was calculated using the following Equation (2):

The dose–response curves were constructed by plotting the scavenging percentage (Equation 2) against the melanoidins concentration (20–100 μg/mL). The IC₅₀ value (concentration required to scavenge 50 % of the DPPH radicals) was determined by non-linear four-parameter logistic regression. Only curves with R² ≥ 0.98 were accepted. The results were expressed as the equivalent concentration of Trolox solution (μg AAE/mL).

2.5.2 ABTS free radical scavenging activity

… After the incubation, the absorbance was measured at 734 nm. A0 and As were the absorbances of the control (distilled water instead of the sample solution) and sample solutions, respectively. Trolox was used as the positive control. ABTS scavenging activity was calculated according to the following Equation (3):

The dose–response curves were constructed by plotting the scavenging percentage (Equation 3) against melanoidin concentration (0.2–1.0 mg/mL). The IC₅₀ value (concentration required to scavenge 50 % of the ABTS radicals) was determined by non-linear four-parameter logistic regression. Only curves with R² ≥ 0.98 were accepted. The results were expressed as the equivalent concentration of Trolox solution (μmol TE/mL).'

Figures and Tables:

In Figure 2, you need to include a footnote explaining the meaning of the letters used for the statistical analysis.

A: Thanks for your kind suggestion. We have added a footnote to Figure 2 to clarify the meaning of the letters used in the statistical analysis.

Ensure consistency in units and decimal formatting across tables.

A: Thanks for your kind suggestion. We have carefully revised all tables in the manuscript to ensure consistency in units and decimal formatting.

References:

Review formatting for consistency with journal guidelines.

A: Thanks for your kind suggestion. We have carefully reviewed the entire manuscript to ensure adherence to the journal's guidelines.

Double-check all references for accuracy, especially DOIs and page numbers.

A: Thanks for your kind suggestion. We have thoroughly double-checked all references in the manuscript to ensure their accuracy, with a particular focus on DOIs and page numbers.

At last, we once again thank the editor and reviewers for your valuable comments and suggestion, which have greatly improved the quality of our manuscript. We hope the revisions have responded all the comments and suggestions, if you still have any question, please feel free to contact us and give us another chance to revise our manuscript.

Ph. D. Xianli Gao* (Corresponding author)

E-mail: gaoxianli@ujs.edu.cn

Tel/Fax: +86 0511-88780201

Reviewer 2 Report

Comments and Suggestions for Authors

Detailed Points for Improvement

 Keywords: Avoid using keywords that already appear in the title of the manuscript.

Materials and methods / Discussion:

  • Deepen the discussion of differences between soy sauce melanoidins and those from other sources (e.g., coffee, bread) beyond what is already mentioned.
  • Address the practical implications of the findings more explicitly.
  • In the Materials and Methods section, the authors state that the results for the DPPH and ABTS assays were expressed as ascorbic acid equivalents (AAE) for DPPH (µg AAE/mL) and Trolox equivalents (TE) for ABTS (µmol TE/mL). However, in the Results and Discussion section, these values are later presented as percentages of contribution to the antioxidant activity of soy sauce, without a clear connection between the initially reported units and this percentage representation. Moreover, there is a lack of clarity on how the authors transitioned from the absolute values (expressed in AAE or TE) to the relative percentage contributions for each molecular weight fraction. This inconsistency between the description of the methods and the presentation of the results creates confusion about how the data were processed and compared. A clearer explanation of the conversion process, including any formulas or normalization steps used to derive these percentages, would improve the transparency and reproducibility of the findings. Additionally, the authors refer to IC50 values (concentration required to scavenge 50% of free radicals) in the Results and Discussion section, particularly when describing the antioxidant activity of the different melanoidin fractions in the DPPH and ABTS assays. However, there is no mention of IC50 determination or any corresponding methodology in the Materials and Methods section. The methods only describe how antioxidant activities were expressed in terms of ascorbic acid equivalents (AAE) or Trolox equivalents (TE), but do not explain how the IC50 values were calculated, what software or statistical tools were used, or how the dose-response curves were constructed. Therefore, clarification is needed. The authors should explicitly describe in the Methods how IC50 values were determined if they intend to present and discuss these results. Otherwise, including IC50 values in the Results without prior methodological description is inconsistent and undermines the transparency of the study.

Figures and Tables:

  • In Figure 2, you need to include a footnote explaining the meaning of the letters used for the statistical analysis.
  • Ensure consistency in units and decimal formatting across tables.

References:

  • Review formatting for consistency with journal guidelines.
  • Double-check all references for accuracy, especially DOIs and page numbers.
Comments on the Quality of English Language

Language Refinement:

  • Improve sentence structure for clarity.
  • Correct minor grammar issues throughout the text (e.g., article use, subject-verb agreement).
  • Simplify overly complex sentences to improve readability.

Author Response

Response to the editor and reviewers

Dear Reviewers,

We are thankful to the editor and the reviewers for your valuable suggestions on your manuscript, which helped us to improve the quality of our manuscript (foods-3748081). We have carefully revised the manuscript according to the suggestions point by point and addressed all the concerns. All the changes were highlighted in the revised manuscript.

Thank you very much for your hard work on our manuscript.

Sincerely yours,

Xianli Gao

Reviewer 2

Comments:

Detailed Points for Improvement

Keywords: Avoid using keywords that already appear in the title of the manuscript.

A: Thanks for your kind suggestion. We have revised the keywords to avoid repetition with the title in the manuscript. The revised keywords now read:

'Keywords: Maillard reaction; soy sauce melanoidins; molecular weight; bioactivity; antioxidant contributions; functional food'

Materials and methods / Discussion:

Deepen the discussion of differences between soy sauce melanoidins and those from other sources (e.g., coffee, bread) beyond what is already mentioned.

A: Thanks for your kind suggestion. The main difference between the Maillard reaction products of soy sauce and those from other sources (such as coffee and bread) lies in their chemical composition. We have provided more details in Section 3.2.1:

'3.2.1 The main composition of melanoidins

…this is due to the fact that soy sauce is typically fermented from protein-rich raw materials such as soybeans and wheat, which are enzymatically hydrolyzed to produce amino acids and small peptides, serving as precursors for the formation of soy sauce melanoidins [3,32]. This is the reason for the higher proportion of protein in soy sauce melanoidins compared with other foods like coffee and chocolate [13,33].

… Compared to the content of phenolic in coffee, chocolate, and sweet wine [33-34], the phenolics content of soy sauce melanoidins is lower. This is primarily attributed to the fact that grain-derived fermented foods have lower phenolic content, resulting in melanoidins proteins with reduced cross-reactivity between gluten, sugars, and phenolic substance…'

Address the practical implications of the findings more explicitly.

A: Thanks for your kind suggestion. I have refined the practical implications of the research findings in the revised draft and summarized them in the conclusion section:

'4. Conclusions

… This study not only deepened our understanding of the antioxidant potential and mechanisms of soy sauce melanoidins but also provides theoretical support for the regulation of antioxidant active components and the development of high-value-added functional seasonings in the deep processing of fermented food…'

In the Materials and Methods section, the authors state that the results for the DPPH and ABTS assays were expressed as ascorbic acid equivalents (AAE) for DPPH (µg AAE/mL) and Trolox equivalents (TE) for ABTS (µmol TE/mL). However, in the Results and Discussion section, these values are later presented as percentages of contribution to the antioxidant activity of soy sauce, without a clear connection between the initially reported units and this percentage representation. Moreover, there is a lack of clarity on how the authors transitioned from the absolute values (expressed in AAE or TE) to the relative percentage contributions for each molecular weight fraction. This inconsistency between the description of the methods and the presentation of the results creates confusion about how the data were processed and compared. A clearer explanation of the conversion process, including any formulas or normalization steps used to derive these percentages, would improve the transparency and reproducibility of the findings. Additionally, the authors refer to IC50 values (concentration required to scavenge 50% of free radicals) in the Results and Discussion section, particularly when describing the antioxidant activity of the different melanoidin fractions in the DPPH and ABTS assays. However, there is no mention of IC50 determination or any corresponding methodology in the Materials and Methods section. The methods only describe how antioxidant activities were expressed in terms of ascorbic acid equivalents (AAE) or Trolox equivalents (TE), but do not explain how the IC50 values were calculated, what software or statistical tools were used, or how the dose-response curves were constructed. Therefore, clarification is needed. The authors should explicitly describe in the Methods how IC50 values were determined if they intend to present and discuss these results. Otherwise, including IC50 values in the Results without prior methodological description is inconsistent and undermines the transparency of the study.

A: Thanks for your kind suggestion. Firstly, to clarify the contribution ratios of different molecular weight components to the antioxidant activity of soy sauce, we have detailed the calculation process in Section 2.6 and provided a brief explanation below Table 4. Secondly, to clarify the presentation format of the DPPH and ABTS assay results, we have provided detailed explanations regarding the calculation of IC50 values in the methods section of Sections 2.5.1 and 2.5.2.

'2.6 Contributions of melanoidins to the antioxidant activities of soy sauce

... Antioxidant activities of the melanoidins fraction solution and soy sauce were assessed using the methods described in Section 2.5 (DPPH、FRAP、ABTS 、 ORAC and MCA). The percentage contribution of each melanoidins fraction to the overall antioxidant activity of soy sauce was calculated using Equation (4):

where Mi is the antioxidant activities of melanoidins fractioni solution (i = 1–3 kDa, 3–10 kDa, 10–30 kDa, 30–50 kDa, and >50 kDa); N is the antioxidant activity of soy sauce measured under the same measurement conditions, Wi is the percentage contribution of fractioni to the antioxidant activity of soy sauce.'

Table 4. Contribution of melanoidins fractions with different molecular weights to antioxidant activities of soy sauce.

Antioxidant Assays

Melanoidins Molecular Weight Fraction Solution

soy sauce

1−3 kDa

3−10 kDa

10−30 kDa

30−50 kDa

>50 kDa

DPPH (μg AAE/mL)

0.81 ± 0.04e

1.47 ± 0.05c

0.91 ± 0.06e

1.17 ± 0.03d

1.80 ± 0.05b

9.25 ± 0.26a

ABTS (μmol TE/mL)

0.04 ± 0.01c

0.05 ± 0.01b,c

0.03± 0.01c

0.05 ± 0.01b,c

0.07 ± 0.01b

0.52 ± 0.03a

OARC (μmol TE/mL)

0.71 ± 0.06d

0.89 ± 0.08c,d

0.77 ± 0.09d

1.13 ± 0.08b,c

1.38 ± 0.09b

8.34 ± 0.28a

MCA (μg EE/mL)

0.78 ± 0.09d

2.08 ± 0.14c

0.96 ± 0.10d

2.30 ± 0.17c

3.17 ± 0.21b

12.38 ± 0.35a

FRAP (μg AAE/mL)

0.92 ± 0.04e

2.94 ± 0.16c

1.51 ± 0.05d

3.37 ± 0.17c

4.42 ± 0.21b

38.46 ± 1.15a

Contribution to DPPH in soy sauce (%)

8.56 ± 0.46e

15.89 ± 0.11b

9.84 ± 0.06d

12.64 ± 0.13c

19.46 ± 0.28a

Contribution to ABTS in soy sauce (%)

7.69 ± 0.37c

9.62 ± 0.41b

5.77 ± 0.26d

9.62 ± 0.35b

13.46 ± 0.30a

Contribution to ORAC in soy sauce (%)

8.51 ± 0.53d

10.67 ± 0.50c

9.23 ± 0.49d

13.55 ± 0.62b

16.55 ± 0.68a

Contribution to MCA in soy sauce (%)

6.30 ± 0.26e

16.80 ± 0.49c

7.74 ± 0.42d

18.58 ± 0.51b

25.61 ± 0.45a

Contribution to FRAP in soy sauce (%)

2.39 ± 0.11e

7.64 ± 0.14c

3.93 ± 0.09d

8.76 ± 0.08b

11.49 ± 0.12a

The antioxidant activities (Mᵢ) are first normalised per milliliter of soy sauce (μg AAE, μmol TE, or μg EE) and then converted to percentage contributions (Wᵢ) using Eq. (4). Values are means ± SD (n = 3). a-e Different letters in the same row indicate the significant differences (p < 0.05).

2.5.1 DPPH free radical scavenging activity

…The mixture was incubated in dark for 30 min, after which the absorbance was measured at 517 nm. A0, A1 and AS are the absorbances of the blank (sample solution was replaced with distilled water), sample and control solutions (DPPH solution was replaced with ethanol), respectively. Ascorbic acid was used as the positive control. DPPH scavenging activity was calculated using the following Equation (2):

The dose–response curves were constructed by plotting the scavenging percentage (Equation 2) against the melanoidins concentration (20–100 μg/mL). The IC₅₀ value (concentration required to scavenge 50 % of the DPPH radicals) was determined by non-linear four-parameter logistic regression. Only curves with R² ≥ 0.98 were accepted. The results were expressed as the equivalent concentration of Trolox solution (μg AAE/mL).

2.5.2 ABTS free radical scavenging activity

… After the incubation, the absorbance was measured at 734 nm. A0 and As were the absorbances of the control (distilled water instead of the sample solution) and sample solutions, respectively. Trolox was used as the positive control. ABTS scavenging activity was calculated according to the following Equation (3):

The dose–response curves were constructed by plotting the scavenging percentage (Equation 3) against melanoidin concentration (0.2–1.0 mg/mL). The IC₅₀ value (concentration required to scavenge 50 % of the ABTS radicals) was determined by non-linear four-parameter logistic regression. Only curves with R² ≥ 0.98 were accepted. The results were expressed as the equivalent concentration of Trolox solution (μmol TE/mL).'

Figures and Tables:

In Figure 2, you need to include a footnote explaining the meaning of the letters used for the statistical analysis.

A: Thanks for your kind suggestion. We have added a footnote to Figure 2 to clarify the meaning of the letters used in the statistical analysis.

Ensure consistency in units and decimal formatting across tables.

A: Thanks for your kind suggestion. We have carefully revised all tables in the manuscript to ensure consistency in units and decimal formatting.

References:

Review formatting for consistency with journal guidelines.

A: Thanks for your kind suggestion. We have carefully reviewed the entire manuscript to ensure adherence to the journal's guidelines.

Double-check all references for accuracy, especially DOIs and page numbers.

A: Thanks for your kind suggestion. We have thoroughly double-checked all references in the manuscript to ensure their accuracy, with a particular focus on DOIs and page numbers.

At last, we once again thank the editor and reviewers for your valuable comments and suggestion, which have greatly improved the quality of our manuscript. We hope the revisions have responded all the comments and suggestions, if you still have any question, please feel free to contact us and give us another chance to revise our manuscript.

Ph. D. Xianli Gao* (Corresponding author)

E-mail: gaoxianli@ujs.edu.cn

Tel/Fax: +86 0511-88780201

Round 2

Reviewer 1 Report

Comments and Suggestions for Authors

I have no additional comments on the manuscript. In my opinion, it is suitable for publication.